# Electrodeposition of Graphene Oxide Modified Composite Coatings Based on Nickel-Chromium Alloy

Vitaly Tseluikin *, Asel Dzhumieva, Andrey Yakovlev, Anton Mostovoy and Marina Lopukhova

Engels Technological Institute, Yuri Gagarin State Technical University of Saratov, Polytechnichskaya St., 77, 410054 Saratov, Russia; aselka2796@gmail.com (A.D.); aw_71@mail.ru (A.Y.); Mostovoy19@rambler.ru (A.M.); mlopuhova@yandex.ru (M.L.)
* Correspondence: tseluikin@mail.ru

**Abstract:** Composite electrochemical coatings (CECs) based on nickel-chromium alloy and modified with multilayer graphene oxide (GO) were obtained. The electrodeposition process of these coatings was studied in the potentiodynamic mode. The structure and the composition of nickel–chromium–GO CECs were studied by scanning electron microscopy and laser microspectral analysis. Nickel–chromium–GO CECs are dense and uniform. The carbon content in them increases when moving from the substrate to the surface. It was established that the addition of GO particles into the composition of electrolytic coatings with a nickel-chromium alloy results in the increase in their microhardness from 4423–5480 MPa to 6120–7320 MPa depending on the cathodic current density.

**Keywords:** electrodeposition; nickel-chromium alloy; graphene oxide; structure; adhesion; microhardness

## 1. Introduction

The deposition of composite electrochemical coatings (CECs) is a reliable and economically feasible method for modifying metal surfaces in order to impart new functional properties to them [1–3]. Among the CECs, coatings based on nickel [4–12] and its alloys [13–22] have become widespread, which is due to the ability of nickel to form electrolytic deposits with dispersed particles of different nature which have good adhesion to the metal base [4]. It should be noted that the electrochemical deposition of alloys is one of the special cases of parallel electrode reactions with their significant mutual influence. Obtaining electroplated coatings with alloys is a technically more complicated process compared to the cathodic deposition of individual metals. However, electrolytic alloys tend to perform better than their individual components. In particular, electrolytic nickel-chromium alloys which are used as hard and wear-resistant coatings [14,21]. It should be noted that the physicomechanical properties (microhardness, wear resistance, etc.) of such coatings depends on the parameters of electrolytic deposition process.

The effectiveness of the practical application of CECs is largely determined by the nature and properties of the dispersed phase. Currently, composite coatings modified with various carbon materials are widely studied: nanodiamonds [9], fullerenes [4], carbon nanotubes [10], carbides [8,12], etc. Graphite and its derivatives are of particular interest as dispersed phases used in the preparation of CECs. Graphite has a pronounced layered structure. In its layers, each carbon atom is bonded to three other atoms 0.142 nm apart. The layers of graphite (graphenes) are arranged in such a way that half of the atoms of one layer are under the centers of the hexagons of the other, and the other half of atoms are under each other. The distance between the layers is 0.335 nm, which is significantly greater than the distance between carbon atoms within one layer. The graphite layer can act both as an electron acceptor, interacting with strong reducing agents, and as an electron donor in reactions with oxidants. When graphite interacts with strong inorganic acids (for example, $H_2SO_4$), graphene oxide (GO) is being formed, which is graphene layers



with oxygen-containing functional groups (hydroxyl, epoxy, carbonyl, etc.) [23]. Graphene and graphene oxide are being extensively explored due to their remarkable operational properties (physicomechanical, electrical, thermal, etc.). In particular, it is shown that the inclusion of graphene in the metal matrix contributes to an increase in the microhardness of the formed coatings [24–26]. However, data of the effect of graphene oxide on the structure and microhardness of electrochemical nickel-chromium alloys are limited.

Thus, the purpose of this work is to obtain nickel–chromium–GO CECs, to investigate the process of their electrodeposition, the structure, and physico-mechanical properties of these coatings.

## 2. Materials and Methods

Nickel–chromium–GO composite coatings were deposited on a steel base (steel 45) from the electrolyte the composition of which is shown in Table 1.

**Table 1.** Electrolyte bath composition and deposition parameters used for nickel–chromium–GO composite coatings.

| № | Electrolyte Composition | Concentration | Deposition Parameters |
|---|---|---|---|
| 1 | $NiSO_4 \cdot 7H_2O$ | 30 g/l | Temperature t = 50 °C |
| 2 | $Cr_2(SO_4)_3 \cdot 6H_2O$ | 150 g/l | Cathodic current density |
| 3 | $H_3BO_3$ | 20 g/l | $i_C$ = 10, 20, 30 A/dm$^2$ |
| 4 | $(NH_4)_2SO_4$ | 40 g/l | |
| 5 | Graphene oxide | 10 g/l | |

Multilayer graphene oxide was added into the electrolyte bath as a powder with a particle size not exceeding 10 micrometers. The process of CECs deposition was carried out with constant stirring of the electrolyte. A pure nickel-chromium alloy was obtained from the above mentioned electrolyte bath without a dispersed phase of GO. The thickness of studied coatings was 40 micrometers (from the outer coating interface to the coating/substrate interface).

Multilayer graphene oxide was synthesized electrochemically in the galvanostatic mode by anodic oxidation of natural graphite powder GB/T 3518-95 (China) with the electricity supply of 700 Ah/kg. 83% $H_2SO_4$ (high purity grade) served as an electrolyte. A detailed description of the procedure for the synthesis of multilayer graphene oxide and the composition of the resulting products are presented in [23].

The specific surface area of graphene oxide was determined by the Brunauer–Emmett–Teller (BET) method using the NOVA 2000e analyzer (Quantachrome Instruments, USA).

The strength of adhesion to the substrate of nickel-chromium coatings was tested by applying a grid of scratches. Several parallel lines with a depth to the substrate at a distance of 2–3 mm from each other and perpendicular to them were made on the surface of the studied coating with a steel-tipped tool at an angle of 30°. Scratches were applied at a constant load on the tool. Adhesion is considered to be satisfactory if the coating does not peel off the metal substrate.

Vickers microhardness (HV) of electrolytic deposits was measured using a PMT-3 device (LOMO, Russia). A tetrahedral diamond pyramid was statically pressed into the studied nickel-chromium coatings under a load of 100 g. The penetration depth of the indenter was 3–4 micrometers. The distance between the indentations was at least two diagonals. The shape of the indentation was square. Taking into account the results of the conducted tests, the values of both diagonals of the indentation were determined. The calculation of the HV values was carried out according to the data of five parallel experiments. The measurement error of device was 3%.

Structural studies were carried out using a scanning electron microscope with a built-in energy dispersive analyzer EXplorer (Aspex, USA).

The composition of the composite coatings was studied using laser microspectral analysis [27]. We used a laser spectroanalytical complex including a Nd: YAG laser

(wavelength 1.06 μm) operating in the giant pulse mode (pulse duration 9 ns). The pulse repetition rate was 25 Hz. The registration system was a DFS-458S diffraction spectrograph and a MIRS attachment (Russia), which included a block of eight receivers, a computer interface board and SPEKTRAN 8 software, which enabled us to carry out qualitative and quantitative spectral analyses, as well as mathematical data processing.

Electrochemical measurements were performed on a P–30J pulse potentiostat (Elins, Russia). The potentials were set relative to a saturated silver chloride reference electrode and recalculated using a standard hydrogen electrode (SHE).

## 3. Results and Discussion

The polarization curves of the deposition of the nickel–chromium alloy and composite coatings on its base in the potentiodynamic mode show that the addition of graphene oxide into the alloying electrolyte facilitates the cathodic process (Figure 1). When the polarization curve is shifted to the negative potentials, superpolarization occurs during the deposition of a metal or alloy. In the opposite case, they talk about depolarization. The nickel-chromium alloy in the presence of dispersed particles is deposited on the cathode at less negative potentials, i.e., the process proceeds with depolarization. The electrodeposition currents of nickel–chromium–GO CECs increase as compared to the coatings with a pure alloy, and this testifies about an increase in the rate of the cathodic process.

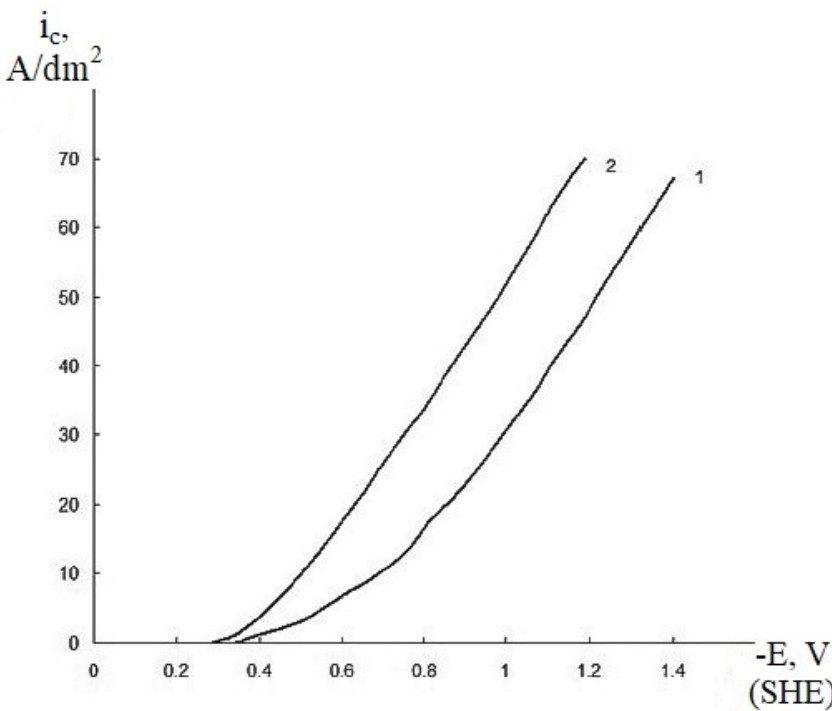

**Figure 1.** Potentiodynamic polarization curves of deposition of nickel–chromium alloy: 1—without additive; 2—together with graphene oxide (potential sweep rate Vs = 8 mV/s).

The study of graphene oxide by scanning electron microscopy (SEM) has shown that it has a layered structure with a developed surface (Figure 2a,b). The specific surface area of graphene oxide determined by the Brunauer–Emmett–Teller (BET) method is 46.78 m$^2$/g. The adsorption of cations from the electrolyte solution can occur on the GO particles, which leads to the formation of a positive charge of the dispersed phase. Therefore, the transfer of GO to the cathode happens probably not only because of the convection, but also due to the action of electrophoretic forces. Cations adsorbed on the GO particles participate in their "bridge" binding with the electrode surface. This binding weakens the disjoining pressure of the liquid layer between the graphene oxide and the cathode, enhancing adhesion [2].

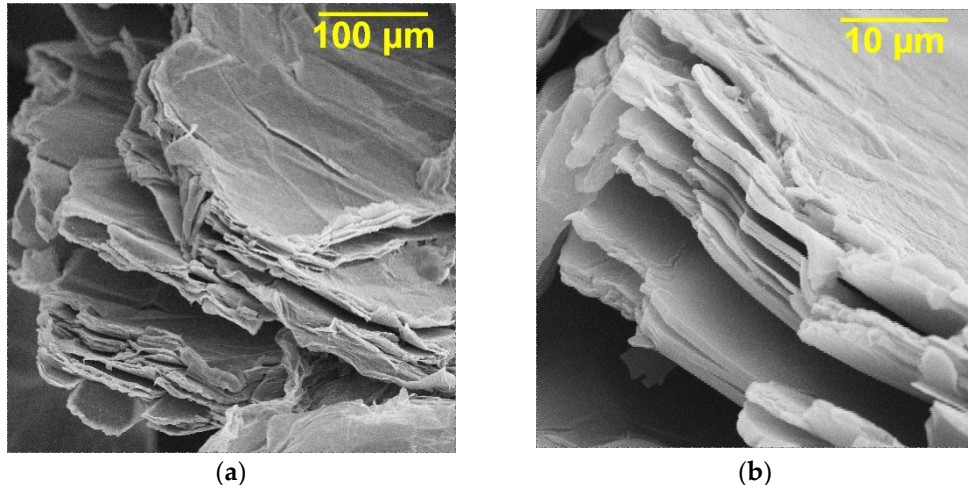

|       |       |
| :---: | :---: |
| (**a**) | (**b**) |

**Figure 2.** SEM images of the graphene oxide structure. Magnification ×500 (**a**), ×5000 (**b**).

The metal overgrowth of the cathode is due to the adsorption forces of the dispersed phase to its surface. This adsorption is carried out step by step. Initially, there is "weak" adsorption between the cathode and dispersed particles fixed on it, which has a physical nature. Particles on the electrode surface are coated with adsorbed metal ions. "Strong" adsorption is irreversible and specific. The particles of the dispersed phase lose their ionic and solvation shells, firmly fixing themselves on the surface of the growing deposit. Strong adsorption is of electrochemical nature, since at this stage metal ions adsorbed on the surface of dispersed particles are discharged [28].

SEM images show that the transition from a pure nickel–chromium alloy (Figure 3a) to a nickel–chromium–GO CEC (Figure 3b) noticeably changes the surface microtopography. The composite coating has an ordered fine-grained structure as compared to a pure alloy. The CEC is dense and uniform, whereas on a nickel–chromium alloy without a dispersed phase, microcracks are observed. Probably, graphene oxide particles on the cathode surface act as crystallization centers, determining the formation and further growth of the electrolytic deposit. It should be noted that nickel–chromium–GO CECs are dense and uniform. When applying a grid of scratches, it was found that all the control samples (both pure nickel–chromium alloys and CECs) did not peel off the coatings from the metal base (Figure 4). The addition of graphene oxide in the nickel–chromium matrix did not worsen the adhesion of the forming coatings, regardless of the current density at which they were deposited.

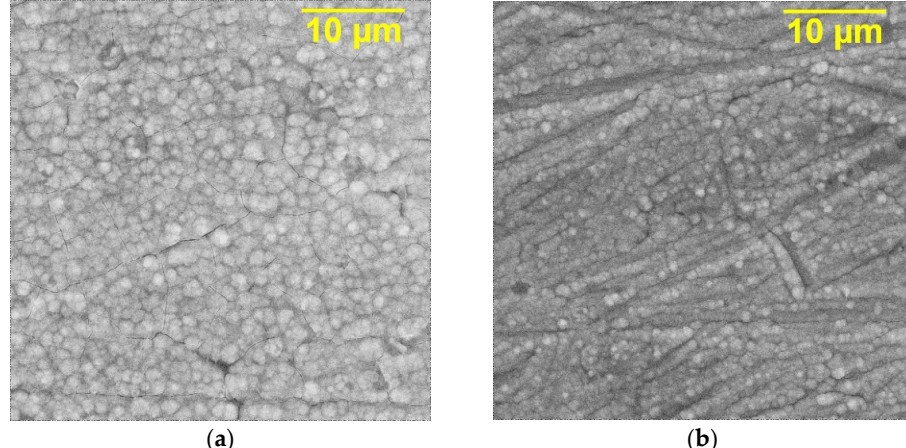

|       |       |
| :---: | :---: |
| (**a**) | (**b**) |

**Figure 3.** SEM images of the surface of the nickel–chromium alloy (**a**) and the CEC nickel–chromium–GO (**b**) deposited at cathodic current density $i_C$ = 10 A/dm$^2$. Magnification ×5000.

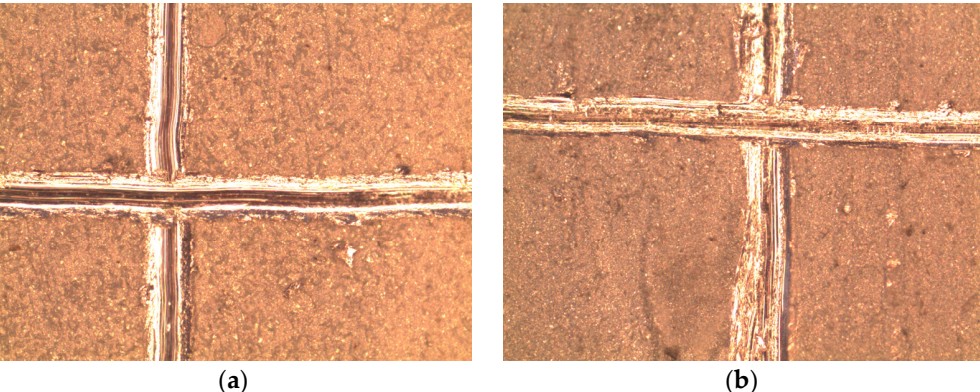

|  |  |
|:---:|:---:|
| (**a**) | (**b**) |

**Figure 4.** The scratches on the surface of the nickel–chromium alloy (**a**) and the CEC nickel–chromium–GO (**b**) deposited at cathodic current density $i_C$ = 10 A/dm$^2$. Magnification ×100.

The addition of a dispersed phase into a galvanic coating results in the change of not only its composition and structure, but also of its operational properties. In practical terms, the physical and mechanical characteristics of metal surfaces, in particular their microhardness, are of significant interest. The study of nickel–chromium–GO CECs by laser microspectral analysis showed that the carbon content in them increases when moving from the substrate to the surface (Figure 5). Besides, the chromium content in the surface layers of the studied composite coatings increases, which should affect their hardness. In out study, with an increase in the cathode current density, an increase in the microhardness of nickel–chromium alloys was observed (Table 2). This is probably due to the inclusion of hydrogen and hydroxides into the deposit, leading to deformation and compression of the coating crystals [21]. As noted above, the addition of a dispersed phase of graphene oxide in nickel–chromium deposits led to their compaction and the formation of fine-crystalline coatings whereas on a pure nickel–chromium alloy microcracks were observed (Figure 3a,b). Therefore, there was an increase in the microhardness of the nickel–chromium–GO CECs in comparison with pure alloys (Table 2), regardless of the electrolysis mode. It is known [1,2] that an important role in the enhancement of the microhardness of the CECs is played by the straightening of the crystallographic orientation and crystalline refinement. Thus, we can assume that during electrochemical deposition of the nickel–chromium–GO composite coatings graphene oxide particles brought about the crystalline refinement and decreased texture of formed deposits.

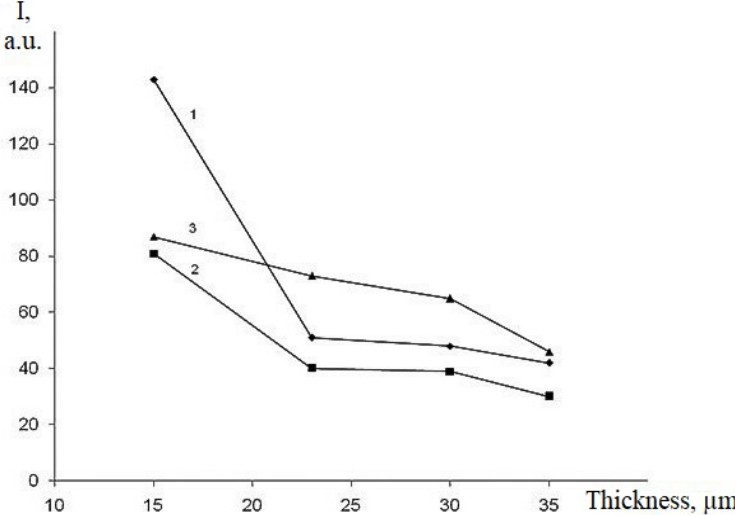

**Figure 5.** Profiles of the concentration of nickel (1), chromium (2) and carbon (3) in the nickel–chromium–GO composite coating obtained at cathodic current density $i_C$ = 10 A/dm$^2$.

**Table 2.** Influence of cathodic current density on the microhardness $HV_{0.1}$, MPa of nickel–chromium alloys and CEC nickel–chromium–GO.

| Cathodic Current Density $i_C$, A/dm$^2$ | $HV_{0.1}$, MPa | |
|---|---|---|
| | Nickel-Chromium | Nickel-Chromium-GO |
| 10 | 4423 | 6120 |
| 20 | 4935 | 6590 |
| 30 | 5480 | 7320 |

Brittleness is an undesirable effect in electrochemical deposits, and its occurrence depends on technological parameters. It is probable that the inclusion of multilayer graphene oxide into the coatings avoids their embrittlement. GO particles on the cathode surface act as crystallization centers, allowing a uniform distribution of the formed coating over the surface and reducing internal stresses during the deposition process. The results of nickel–chromium alloys microhardness are consistent with other literature data [14,21,25,26].

## 4. Conclusions

On the basis of the conducted studies, it was found that addition of a dispersion of multilayer graphene oxide into the deposition electrolyte of the nickel-chromium alloy results in the formation of composite electrochemical coatings. The inclusion of GO particles into the composition of nickel-chromium deposits leads to a change in the structure of their surface and physical and mechanical properties. Nickel–chromium–GO CECs are dense and uniform. The addition of graphene oxide in the nickel–chromium matrix does not worsen the adhesion of the forming coatings. The microhardness values of the studied CECs increase with an increase in the cathode current density from 6120 MPa to 7320 MPa.

**Author Contributions:** Conceptualization, V.T.; Data curation, A.Y.; Formal analysis, A.Y. and A.M.; Investigation, V.T. and A.D.; Methodology, V.T., A.D., A.M. and A.Y.; Project administration, V.T.; Supervision, V.T.; Visualization, A.D., A.Y., A.M. and M.L.; Writing—original draft, V.T.; Writing—review & editing, V.T., A.D., A.Y., A.M. and M.L. All authors participated in the discussion of the results and the writing of the text of the article. All authors have read and agreed to the published version of the manuscript.

**Funding:** The reported study was funded by RFBR according to the research project № 18-29-19048.

**Institutional Review Board Statement:** Not applicable.

**Informed Consent Statement:** Not applicable.

**Conflicts of Interest:** The authors declare no conflict of financial and non-financial interest requiring disclosure in this article.

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
