# Peer review of "Electrodeposition of Graphene Oxide Modified Composite Coatings Based on Nickel-Chromium Alloy"

_crystals, doi:10.3390/cryst11040415_

Round 1
Reviewer 1 Report
The authors propose the eletrodeposited Ni-Cr alloys with GOs. The purpose of this study is not clear and persuasive, and the organization is like a experimental report. The properties of Ni-Cr alloy must be checked, including microstructure and chemical composition. The SEM images present the stacking of layered carbon material, still graphite. In addition, no further discussion in the relationship of electrodepositing conditions and hardness.
Author Response
In the new version, the paper is supplemented and corrected.
Reviewer 2 Report
Herein, the Authors report the deposition of the composite coatings based on nickel-chromium alloy and modified with multilayer graphene oxide. The coatings were electrochemically deposited from an electrolyte that contained graphene oxide, on steel substrates. The deposition mechanism of the coatings has been described and some structural and micromechanical investigations have been done. The microhardness and the degree of coating adherence to the substrate were determined. Unfortunately, in many places both the methodology and the research results have not been clearly explained.
This work is not at the correct standard to be published at this stage and requires major corrections before it is suitable for publication (see comments later).

Author Response
Authors Notes to Reviewer in the Attached File

Reviewer 3 Report
The paper describes preparation of an electrodeposited Ni-Cr coating with incorporated graphene oxide particles, shows its surface structure and gives microhardness values. Although it would be much more informative to a reader if the coating is characterized in detail, including the microstructure, tribological properties, corrosion resistance, etc., as a function of deposition parameters, it is still worth publishing. However, there are several critical shortcomings in view of the clarity and quality of scientific writing that need to be addressed prior to the acceptance:
- In the Abstract, it is claimed that „the electrodeposition process of these coatings has been studied in the potentiodynamic mode.“ In Table 1, cathodic current densities are shown, indicating the deposition was performed in a potentiostatic mode. Can you clarify this in the experimental part?
- Lines 135-137. „According to the data of laser microspectral analysis, with an increase in the cathode current density the studied alloys are enriched with chromium, which also contributes to an increase in their microhardness.“ No data supproting this statement are given. Without presenting them, it is not possible to keep such a statement in the paper.
- Lines 130-131. „Besides, the chromium content in the surface layers of the studied coatings increases, which should affect their hardness.“ Only data on the chromium content in the coating containing graphene oxide particles are shown. It is no possible to conclude the change in the micro-hardness is linked to the chromium content. It is necessary to provide data for the reference coating as well. Otherwise, this statement is only a speculation.
- Lines 133-134. „This is probably due to the addition of hydrogen and hydroxides into the deposit, leading to deformation and compression of the coating crystals.“ Can you provide any support for this hypothesis? Any data, or reference to previous literature? It was not even shown the coating is crystalline.
- Lines 103-110. It is not clear where data for description of the mechanism came from. It should be made obvious if this paragraph is a reference to another paper or if there is any piece of evidence for the described mechanism based on data given in this paper.
Minor comments follow:
- Information about the coating thickness is missing.
- The cross-cut test should probably be performed with the distance of cuts of 1 mm. Larger distance is recommended only for thick coatings, see ISO 2409.
- Line 87. What exactly the authors mean by „the process proceeds with depolarization“? Please, explain.
- Lines 93-100. This paragraph should come before the first one, lines 83-89.
- Line 115. Please, mark the micro-cracks with arrows in the micrograph.
- Figure 4. What is the unit „mcm“? What the thickness refers to? From what point? This needs to be better explained.
Author Response
- In the Abstract, it is claimed that „the electrodeposition process of these coatings has been studied in the potentiodynamic mode.“ In Table 1, cathodic current densities are shown, indicating the deposition was performed in a potentiostatic mode. Can you clarify this in the experimental part?
Results of potentiodynamic study shows Figure 1.
- Lines 135-137. „According to the data of laser microspectral analysis, with an increase in the cathode current density the studied alloys are enriched with chromium, which also contributes to an increase in their microhardness.“ No data supproting this statement are given. Without presenting them, it is not possible to keep such a statement in the paper.
This statement is removed from the text of the paper.
- Lines 130-131. „Besides, the chromium content in the surface layers of the studied coatings increases, which should affect their hardness.“ Only data on the chromium content in the coating containing graphene oxide particles are shown. It is no possible to conclude the change in the micro-hardness is linked to the chromium content. It is necessary to provide data for the reference coating as well. Otherwise, this statement is only a speculation.
It was meant to increase the chromium content in composite coatings. In the new version of the paper, the phrase reads as follows: „Besides, the chromium content in the surface layers of the studied composite coatings increases, which should affect their hardness.“ (lines 151 - 153).
- Lines 133-134. „This is probably due to the addition of hydrogen and hydroxides into the deposit, leading to deformation and compression of the coating crystals.“ Can you provide any support for this hypothesis? Any data, or reference to previous literature? It was not even shown the coating is crystalline.
. In the new version of the paper added a reference to previous literature (lines 154 - 156).
- Lines 103-110. It is not clear where data for description of the mechanism came from. It should be made obvious if this paragraph is a reference to another paper or if there is any piece of evidence for the described mechanism based on data given in this paper.
. In the new version of the paper added a reference to another paper (line 126).
Minor comments follow:
- Information about the coating thickness is missing.
This information is added to the text of the article (line 61).
- The cross-cut test should probably be performed with the distance of cuts of 1 mm. Larger distance is recommended only for thick coatings, see ISO 2409.
The thickness of the studied coatings was 40 microns.
- Line 87. What exactly the authors mean by „the process proceeds with depolarization“? Please, explain.
The new version of the paper explains this statement. “When the polarization curve is shifted to the negative potentials, superpolarization occurs during the deposition of a metal or alloy. In the opposite case, they talk about depolarization” (lines 98 - 100).
- Lines 93-100. This paragraph should come before the first one, lines 83-89.
From our point of view, the paragraphs are in the correct order.
- Line 115. Please, mark the micro-cracks with arrows in the micrograph.
We believe that the arrows on the micrograph will distort its perception. Micro-cracks are visible, especially when magnified.
- Figure 4. What is the unit „mcm“? What the thickness refers to? From what point? This needs to be better explained.
The unit „mcm“ is corrected in the Figure 4. This means the thickness of the coating.
Round 2
Reviewer 1 Report
After reading the revised manuscript, I do not notice any improvement. Since this study focused on the coating's properties, the influence of parameters or the possible reasons in the films of different properties should be discussed, not only report the data. Thus, it is suggested not to consider the publication of it.
Author Response
The authors submit a re-corrected manuscript.
Reviewer 2 Report
The authors responded to the review, but the manuscript was only partially suplemented and developed. The article discusses an interesting issue and therefore should be completed and refined. The following points included in the file should be considered.

Author Response
The authors are grateful to the Reviewer for a detailed review of the manuscript and valuable comments. The authors notes to Reviewer in the attached file.

Reviewer 3 Report
I would like to thank the authors for improving the manuscript following my comments. However, there are still several points that should be re-considered before publication.
- In the Experimental, lines 59-64, it is not explained that the electrodeposition process was performed under galvanostatic conditions at current density of 10, 20, and 30 A/dm2 (Table 2). Please, state it explicitly; currently, readers can misinterpret the mention of the potentiodynamic study in the Abstract and elsewhere with the deposition method.
- Line 63. Instead of microns, write micrometers. In addition, it should be explained how was the thickness measured and what was the measurement error.
- Figure 1. There is probably a typo. The y axis unit should read A/dm2, not A/sm2.
- Figure 4. It is still not clear what the thickness values refer to. It should be explained that 0 was the outer coating interface and 40 (?) the coating/substrate interface.
- Line 152-155. „Besides, the chromium content in the surface layers of the studied composite coatings increases, which should affect their hardness. Indeed, with an increase in the cathode current density, an increase in the microhardness of nickel–chromium alloys is observed (Table 2).“ The sentences suggest that the increase of the micro-hardness with the current density was proved to be linked to the chromium content in the surface layer. This cannot be stated because no data on the chromium content for the other coatings is given. Please, remove the word „indeed“, to separate the two statesments.
Author Response
In the Experimental, lines 59-64, it is not explained that the electrodeposition process was performed under galvanostatic conditions at current density of 10, 20, and 30 A/dm2 (Table 2). Please, state it explicitly; currently, readers can misinterpret the mention of the potentiodynamic study in the Abstract and elsewhere with the deposition method.
The corresponding addition is made to the text of the paper (Table 1).
Line 63. Instead of microns, write micrometers. In addition, it should be explained how was the thickness measured and what was the measurement error.
The corresponding addition is made to the text of the paper (lines 68 - 69).
Figure 1. There is probably a typo. The y axis unit should read A/dm2, not A/sm2.
There is not a typo. The y axis unit is i * 103, A/sm2.
Figure 4. It is still not clear what the thickness values refer to. It should be explained that 0 was the outer coating interface and 40 (?) the coating/substrate interface.
The paper indicates that «the study of nickel–chromium–GO CECs by laser microspectral analysis has shown that the carbon content in them increases when moving from the substrate to the surface» (lines 162 - 164) and it follows from the text that 0 is the outer coating interface and 40 - the coating/substrate interface.
Line 152-155. „Besides, the chromium content in the surface layers of the studied composite coatings increases, which should affect their hardness. Indeed, with an increase in the cathode current density, an increase in the microhardness of nickel–chromium alloys is observed (Table 2).“ The sentences suggest that the increase of the micro-hardness with the current density was proved to be linked to the chromium content in the surface layer. This cannot be stated because no data on the chromium content for the other coatings is given. Please, remove the word „indeed“, to separate the two statesments.
The word „indeed“ is removed from the text of paper.